# SARS-CoV-2 Nsp6 damages *Drosophila* heart and mouse cardiomyocytes through MGA/MAX complex-mediated increased glycolysis

Jun-yi Zhu [1,2], Guanglei Wang[3], Xiaohu Huang[1,2], Hangnoh Lee [1,2], Jin-Gu Lee[1,2], Penghua Yang[3], Joyce van de Leemput [1,2], Weiliang Huang[4,5], Maureen A. Kane[4], Peixin Yang [3] & Zhe Han [1,2 ✉]

SARS-CoV-2 infection causes COVID-19, a severe acute respiratory disease associated with cardiovascular complications including long-term outcomes. The presence of virus in cardiac tissue of patients with COVID-19 suggests this is a direct, rather than secondary, effect of infection. Here, by expressing individual SARS-CoV-2 proteins in the *Drosophila* heart, we demonstrate interaction of virus Nsp6 with host proteins of the MGA/MAX complex (MGA, PCGF6 and TFDP1). Complementing transcriptomic data from the fly heart reveal that this interaction blocks the antagonistic MGA/MAX complex, which shifts the balance towards MYC/MAX and activates glycolysis—with similar findings in mouse cardiomyocytes. Further, the *Nsp6*-induced glycolysis disrupts cardiac mitochondrial function, known to increase reactive oxygen species (ROS) in heart failure; this could explain COVID-19-associated cardiac pathology. Inhibiting the glycolysis pathway by 2-deoxy-D-glucose (2DG) treatment attenuates the *Nsp6*-induced cardiac phenotype in flies and mice. These findings point to glycolysis as a potential pharmacological target for treating COVID-19-associated heart failure.

[1] Center for Precision Disease Modeling, Department of Medicine, University of Maryland School of Medicine, 670 West Baltimore Street, Baltimore, MD 21201, USA. [2] Division of Endocrinology, Diabetes and Nutrition, Department of Medicine, University of Maryland School of Medicine, 670 West Baltimore Street, Baltimore, MD 21201, USA. [3] Department of Obstetrics, Gynecology & Reproductive Sciences, University of Maryland School of Medicine, Baltimore, MD 21201, USA. [4] Department of Pharmaceutical Sciences, University of Maryland School of Pharmacy, Baltimore, MD 21201, USA. [5] Present address: University of Queensland, Brisbane, QLD 4072, Australia. ✉email: zhan@som.umaryland.edu

COVID-19, the disease caused by SARS-CoV-2 infection, is notable for its wide range in severity; from asymptomatic to severe acute respiratory syndrome, systemwide organ failure, and ultimately death. Even though COVID-19 is known as a severe acute respiratory syndrome (SARS), it presents with a wide variety of symptoms that often include cardiac pathology. The detection of SARS-CoV-2 in cardiac tissue[1–7], suggests the cardiac pathology is caused by direct virus action, rather than secondary complications. Notably, elevated levels of cardiac bio-markers—such as troponins, myoglobin, C-reactive protein, interleukins and natriuretic peptides—indicative of myocardial injury, have been observed in COVID-19[8]. Their presence, combined with abnormal echocardiograms that reflect functional deficits, has been associated with a poorer prognosis of disease progression[9–11]. Indeed, cardiovascular complications reported in patients with COVID-19 include necrosis, ventricular dysfunction, heart failure, and arrhythmia[8]. Moreover, there is evidence of myocardial inflammation (myocarditis) following SARS-CoV-2 infection, including in asymptomatic individuals[12,13]; and, the long-term (1-year) risk of cardiovascular disease is considerable in patients with COVID-19, independent of hospitalization[14].

The key to treating the SARS-CoV-2 induced cardiovascular pathology likely lies in understanding and targeting the individual virus-host interactions[15]. Initially these have been investigated in large-scale network studies which revealed distinct proteome interaction networks, for many of these interactions specific pharmacological compounds exist[16–22]. Of these compounds, numerous are currently being tested in clinical trials for their efficacy in treating COVID-19. More recently, studies have been published that delve deeper into the individual SARS-CoV-2 proteins, their host interactions and underlying pathomechanisms. These have already revealed that some virus proteins known for their role in virus replication processes, such as SARS-CoV-2 Nsp3 and Nsp5 which encode the (PLpro) and (3CL pro; main protease, Mpro), are also responsible for disrupting host immune signaling processes through specific protein-protein interactions[23,24]. These findings suggest that therapeutics targeting these virus proteins specifically could both reduce virus replication and diminish their role in the damaging host pathogenic effects.

Viruses often take control of host metabolic processes to support virus replication[25,26]. These metabolic and signaling pathways are highly conserved, and studies in Drosophila have led to the discovery of several of these systems, preceding their identification in mammalian species[27]. As such, the fly has proven valuable in the study of virus-host interaction and related pathogenic mechanisms for a variety of human viruses, including human immunodeficiency virus (HIV), Zika virus, and dengue virus, to name but a few[28–30]; and, the fly has been used to study specific coronavirus proteins and their pathogenic effects on host cells. For example, overexpression of SARS-CoV Orf3a or membrane (M) proteins caused cytotoxicity in the fly eye[31,32]. Moreover, Drosophila forward genetic screens were used to identify host genetic modifiers of these pathogenic virus-host protein interactions. Notably, nearly 60% of the modifiers identified in these fly studies have human orthologs that are expressed in lung tissue[31,32]. And a recent publication by our group identified Orf6, Nsp6 and Orf7a to be the most pathogenic SARS-CoV-2 proteins when expressed in flies[33]. Further study revealed damage to muscle, characterized by reduced mitochondria, and abnormal trachea (fly equivalent of human lung). Interestingly, treatment with selinexor, a compound known to target XPO1, the human host protein interacting with SARS-CoV-2 Orf6 protein, successfully attenuated the Orf6-induced phenotypes. However, selinexor was unable to treat the highly similar phenotypes induced by SARS-CoV-2 Orf7a or Nsp6,

suggesting the underlying pathomechanism is specific to each virus protein[33]. The relevance of these studies is evident in the extremely high level of conservation of the virus-host inter-actome, with 90% of human proteins in the SARS-CoV-2 protein interaction network conserved from flies[33,34].

Altogether, the fruit fly makes a remarkably powerful and versatile model system. It combines the speed and versatility of in vitro models with the whole-body physiology (access to all major organs and tissues) of in vivo models. Despite the initial obvious differences, at the core flies and mammalian organisms share many molecular and biological systems. As such, flies have been instrumental in studying a variety of human diseases including developmental (congenital heart disease) and metabolic (diabetic cardiomyopathy) heart disorders. This is made possible by the Drosophila heart tube resembling the earliest stages of mammalian cardiac development, and due to the high level of conservation of key metabolic processes and regulatory mechanisms, including the lipid and glucose pathways[35]. Conservation of these systems is even observed when studying the effects of age, high fat diet, and the associated increased levels of ROS on heart function

Here we used Drosophila to study the effect of individual SARS-CoV-2 proteins on the heart. Findings revealed that expression of Nsp6 specifically in the fly heart causes pronounced structural and functional damage, in resemblance of the myocardial injury and functional abnormalities observed in patients with COVID-19. Furthermore, the data show this is mediated by direct interaction of Nsp6 with the MGA/PCGF6/TFDP1/MAX [polycomb repressive complex 1.6 (PRC1.6)] regulatory complex for glycolysis activity. Data from mouse cardiomyocytes confirmed the SARS-CoV-2 Nsp6-induced glycolysis and cardiac distress. Through this host interaction, Nsp6 shifts the balance towards MYC/MAX regulation, thus leading to increased glycolysis and the observed heart phenotype. These findings support a role for dysregulated glycolysis in COVID-19, specifically in heart pathology. Inhibition of glycolysis by treatment with 2-deoxy-D-glucose (2DG) largely attenuated the Nsp6-induced heart phenotype in flies and mice. Thus, identifying the glycolysis pathways as a therapeutic target for SARS-CoV-2 instigated heart failure in COVID-19 patients.

## Results

**Heart-specific expression of individual SARS-CoV-2 transgenes causes developmental lethality and a reduced lifespan.** Previously, we have generated transgenic fly lines each expressing an individual SARS-CoV-2 gene encoding one of the 12 virus proteins most likely to instigate pathogenic host interactions[33]. SARS-CoV-2 infection can progress to COVID-19, which is characterized by multiple cardiac manifestations. To understand how SARS-CoV-2 causes heart damage, we decided to use the heart-specific driver 4XHand-Gal4 to directly express these 12 SARS-CoV-2 genes in the Drosophila heart and then study their pathogenic outcomes on cardiac structure and function. It should be noted that typically SARS-CoV-2 infection affects mature cells, whereas in our model system using 4XHand-Gal4, the SARS-CoV-2 genes are expressed from the early embryonic stages with viral protein present throughout fly heart development.

Male and female flies of the designated genotypes were crossed to produce progeny carrying the corresponding UAS-SARS-CoV-2 gene construct, either with heart-specific expression of the individual virus gene driven by 4XHand-Gal4 (straight wing), or with the balancer (CyO, curly wing) (Fig. 1a). We found that the expression of SARS-CoV-2 Nsp, Orf6 or Orf7a in the fly heart induced a high mortality rate, while Nsp3

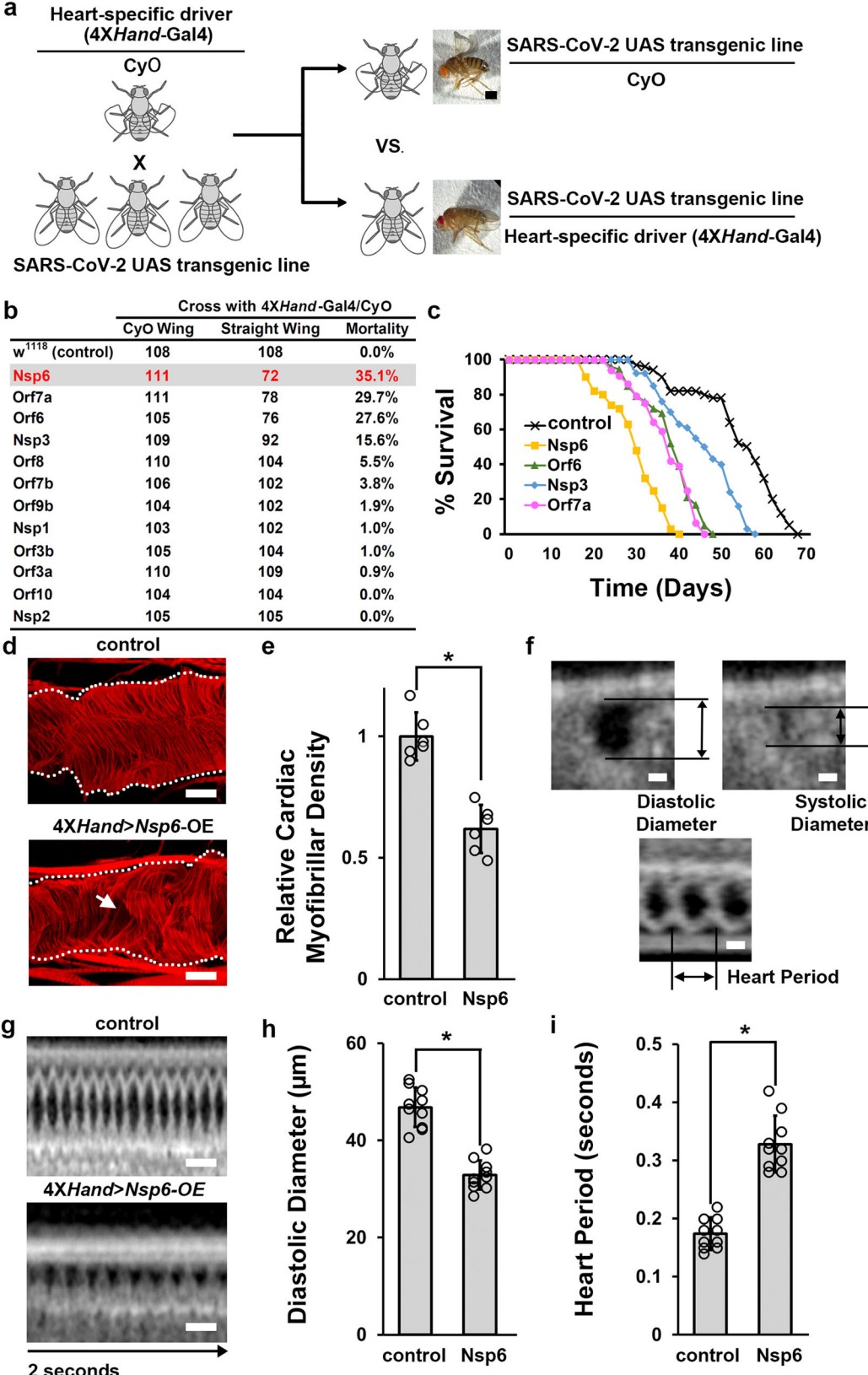

flies showed a mildly increased mortality rate (Fig. 1b; Supplementary Fig. 1). For the adult flies that eclosed, their lifespans were monitored. We found *Nsp6* expression in the fly heart caused a shortened lifespan with all flies dead by 40 days, whereas control flies lived up to 70 days (Fig. 1c). Flies with

cardiac expression of *Orf6* or *Orf7a* showed a shortened lifespan as well, living a maximum of 50 days (Fig. 1c). These results suggest that these SARS-CoV-2 encoded proteins (Nsp6, Orf6, Orf7a and Nsp3) may be associated with cardiac pathology, with Nsp6 having the most damaging effect in our model.

**Fig. 1 SARS-CoV-2 Nsp6, Orf6 or Orf7a transgene expression causes developmental lethality, heart morphological and functional defects. a** Schematic representation of the genetic screen to identify individual SARS-CoV-2 genes with cardiac pathology. Fly images taken using a ZEISS SteREO Discovery.V12 microscope. Scale bar = 0.5 mm. **b** Quantitation of mortality rate prior to eclosion for each individually expressed SARS-CoV-2 gene from the crosses in **a**. Mortality was calculated as (CyO wing − straight wing)/CyO wing × 100. $n$ = ~200 flies (across 4 vials). **c** Graph displaying the lifespan for adult flies carrying SARS-CoV-2 Nsp6, Orf6, Orf7a or Nsp3 transgene expression. $w^{1118}$ is a wild-type control. **d** The adult heart phenotype induced by 4XHand-Gal4, cardioblast-specific overexpression (OE) of the UAS-SARS-CoV-2 Nsp6 transgene. Cardiac actin myofibers were visualized by phalloidin staining. Dotted lines delineate the outline of the heart tube. Arrow points to missing cardiac myofibers. $w^{1118}$ is a wild-type control. Scale bars (both) = 40 μm. **e** Quantitation of adult heart cardiac myofibrillar density relative to control. SARS-CoV-2 Nsp6 (4XHand-Gal4) flies, and $w^{1118}$ is a wild-type control. $P$ value = 5.3E−05. $n$ = 6 flies per genotype. **f** Schematic representation of heart diastolic and systolic diameter, and the heart period by optical coherence tomography (OCT). Scale bars (top two panels) = 20 μm; (bottom) = 30 μm. **g** Drosophila heartbeat video images from flies induced by 4XHand-Gal4 cardioblast-specific overexpression (OE) of the UAS-SARS-CoV-2 Nsp6 transgene. $w^{1118}$ is a wild-type control. Scale bars (both) = 25 μm. **h** Quantitation of adult heart diastolic diameter. $n$ = 10 flies per genotype. $P$ value = 8.3E−08. **i** Quantitation of heart period, i.e., indication of heart rate. $n$ = 10 flies per genotype. $P$ value = 4.0E−08. Statistical significance (*) defined as $P < 0.05$ (Student's $t$ test); data shown as mean ± SD.

**Heart-specific expression of the SARS-CoV-2 Nsp6 transgene causes heart morphological and functional defects**. To better understand the extent of individual SARS-CoV-2 Nsp6-induced heart damage, we first investigated if there were any morphological changes in the Drosophila heart. The fly hearts were stained with phalloidin to visualize the structure of the cardiac actin filaments (myofibrils). Heart-specific expression of Nsp6 caused disorganized cardiac actin filaments and significantly reduced cardiac muscle fiber density (Fig. 1d, e). Next, we applied optical coherence tomography (OCT) to assess any cardiac functional defects induced by heart-specific expression of Nsp6. The cardiac diastolic and systolic diameter, and the heart period were measured to determine cardiac function (Fig. 1f, g). We found that Nsp6 expression reduced the diastolic (Fig. 1g, h)—but not the systolic (Fig. 1g; Supplementary Data 3)—diameter of the heart tube. Furthermore, Nsp6 expression significantly affected the heart period (i.e., reduced heart rate) (Fig. 1i; Supplementary Movie 1 and 2). These results indicate that SARS-CoV-2 Nsp6 can induce structural heart damage and cardiac functional defects.

**Glycolysis genes are upregulated in the SARS-CoV-2 Nsp6 transgenic fly hearts**. To gain insight into the mechanism underlying the cardiac phenotype, we performed RNAseq analysis of the dissected fly hearts that specifically expressed SARS-CoV-2 Nsp6 (Supplementary Data 1). Among the significantly downregulated genes we found those associated with Gene Ontology (GO) terms in the ribosome biogenesis (adj. $P$ = 1.05e −52), and rRNA or ncRNA metabolic processes (adj. $P$ = 7.92e −52 and 1.37e−51, respectively) (Supplementary Fig. 2). Previously SARS-CoV-2 has been shown to bind to the ribosome to disrupt protein translation[36]. Moreover, downregulation of genes associated with ribosome biogenesis has been shown following HIV-1 infection in an immortalized human T lymphocyte cell line[37].

The differential gene expression profile revealed significantly increased expression of carbon metabolism genes (Fig. 2a; adj. $P$ value < 0.05, Wald test, corrected with Benjamini–Hochberg method). For example, Ald gene encodes Aldolase, a key enzyme in glycolysis that converts Fructose 1,6-bisphosphate into two triose phosphates, Dihydroxyacetone phosphate and Glyceraldehyde 3-phosphate[38]; Ald was upregulated 2.68 times in fly hearts with Nsp6 expression (adj. $P$ = 4.84e−13). Similarly, genes encoding proteins upstream of Ald in the glycolytic pathway, such as Hex-C (Hexokinase C, 2.55 fold, adj. $P$ = 4.04e−06) and Pgi (Phosphoglucose isomerase, 2.45 fold, adj. $P$ = 2.23e−08), as well as downstream of it, PyK (Pyruvate kinase, 2.13 fold, adj. $P$ = 1.7e−07), showed increased expression in the Nsp6 expressing fly hearts. In addition, we observed a significant increase of CG32444 expression (6.15 fold, adj. $P$ = 5.11e−20), which encodes an ortholog of human GALM (Galactose mutarotase).

This protein is not a core member of glycolysis, but its function is required for the utilization of galactose via glycolysis[39].

The differential expression of multiple glycolytic genes in response to SARS-CoV-2 Nsp6, could reflect altered steps in cellular respiration or carbon metabolism. To test this, we subdivided the metabolic genes into four groups based on their classification in KEGG (Kyoto Encyclopedia of Genes and Genomes)[40]. Genes involved in glycolysis showed greater upregulation (median = 1.84 fold), than those encoding proteins in the tricarboxylic acid cycle (TCA, median = 1.25 fold, $P$ = 0.0039, Mann–Whitney U test) or oxidative phosphorylation (OxPhos, median = 1.32 fold, $P$ = 7.389e−09) (Fig. 2b, c). Glycolysis is tightly linked to the pentose phosphate pathway (PPP). It has been postulated that enhanced glycolysis and PPP are beneficial for viral replication by providing PPP intermediates for the nucleotide precursors[41]. Consistent with this idea, we found that Nsp6 expression led to PPP gene upregulation (median = 2.01 fold). This upregulation was comparable in scope to the upregulation of the glycolysis genes (median = 1.84 fold) following Nsp6 expression ($P$ = 0.96; i.e., no significant difference between PPP and glycolysis gene upregulation in Nsp6 flies). Interestingly, the Drosophila glycolysis pathway genes are highly conserved from fly to human based on their ortholog conservation scores (DIOPT; Fig. 2c).

Pathway analysis further supported our findings and demonstrated over-representation of carbon metabolism, especially glycolysis-related, genes among the upregulated genes (Fig. 2d, e). GO analysis showed enrichment of functional terms linked to the oxoacid metabolic process (adj. $P$ = 3.94e−21, Hypergeometric test, corrected with Benjamini–Hochberg method), carboxylic acid metabolic process (adj. $P$ = 7.27e−21), and carbohydrate metabolic process (adj. $P$ = 1.46e−24) (Fig. 2d). KEGG pathway analysis provided a deeper understanding of the specific carbon metabolism pathways affected by Nsp6. It showed that genes involved in glycolysis/gluconeogenesis (adj. $P$ = 5.21e−9) and PPP (adj. $P$ = 7.29e−6) were significantly enriched among the upregulated genes (Fig. 2e).

Altogether, these findings indicate that SARS-CoV-2 Nsp6-mediated upregulation of glycolysis might contribute to the observed cardiac defect in flies and given the high level of conservation, possibly in COVID-19 patients as well.

**SARS-CoV-2 Nsp6-induced glycolysis and increased Pgi are associated with heart morphological and functional defects**. The gene expression data indicated that SARS-CoV-2 Nsp6 induced glycolysis. To validate this finding with a biochemical assay, we quantified the activity of Phosphoglucose isomerase (Pgi). Pgi is a key enzyme at the start of the glycolysis pathway that interconverts Glucose-6-phosphate and Fructose-6-phosphate (Fig. 3a). The assay confirmed that ubiquitous

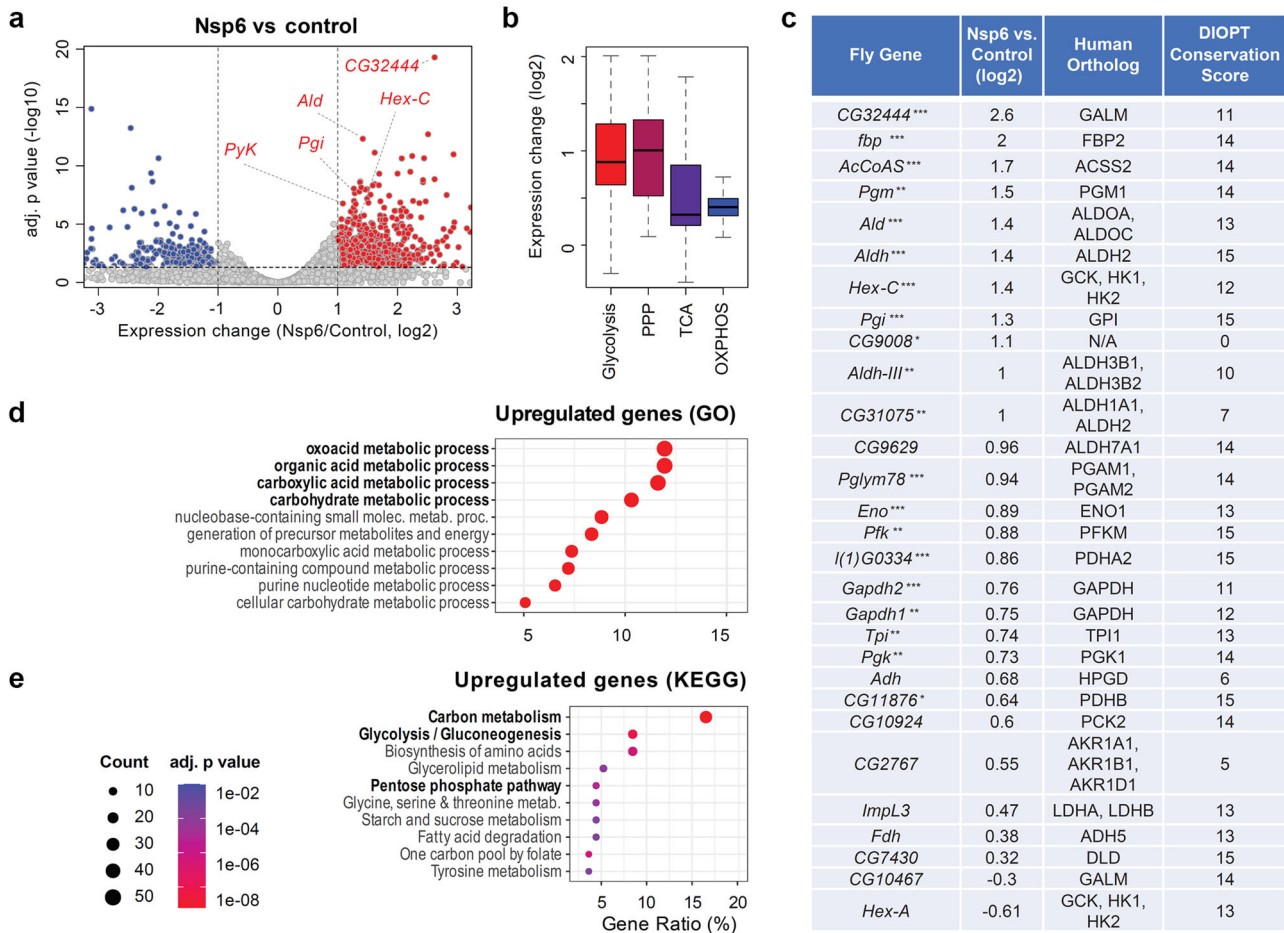

**Fig. 2 SARS-CoV-2 Nsp6 alters the expression of host glycolysis genes in the fly heart. a** Differential expression of fly heart genes upon 4X*Hand*-Gal4 cardioblast-specific transgenic overexpression (OE) of UAS-*SARS-CoV-2 Nsp6*. X-axis: Gene expression differences between the *Nsp6*-expressing heart versus wild-type control ($w^{1118}$) in log2 scale. Y-axis: Adjusted $P$ values in −log10 scale. Each dot represents a gene: Red color denotes genes upregulated in *Nsp6*-OE fly hearts; Blue color denotes genes downregulated in *Nsp6*-OE fly hearts. **b** Box plot summarizes differentially expressed genes in SARS-CoV-2 *Nsp6*-OE fly heart compared to wild-type control ($w^{1118}$) heart, involved in Glycolysis, Pentose Phosphate Pathway (PPP), TriCarboxylic Acid cycle (TCA) and OXidative PHOSphorylation (OXPHOS). The top and bottom of each box corresponds to the upper and lower quartiles, respectively. Each thick line represents the median value. Whiskers indicate the largest, or smallest, observations within 1.5 times of the interquartile range (upper-lower) from the top/bottom of the boxes, respectively. **c** Gene expression changes of fly glycolysis genes with their human orthologs. Orthology information and conservation scores were obtained from DIOPT (max conservation score = 15). * adjusted $P < 0.05$, ** adjusted $P < 0.01$, *** adjusted $P < 0.001$. Enrichment of genes with specific Gene Ontology (GO) terms (**d**) or KEGG-defined pathways (**e**). Circle size: Number of genes associated with the function. Color: Adjusted $P$ values. X-axis represents the percentage of genes with the function among all the significantly upregulated (adj. $P$ value < 0.05) genes.

expression of SARS-CoV-2 *Nsp6* significantly increases Pgi activity (Fig. 3b). In addition, we measured NADH levels, a metabolite produced from NAD+ through glycolysis. Similarly, NADH levels significantly increased following ubiquitous expression of *Nsp6* in flies (Fig. 3c). Combined, the RNAseq gene expression and biochemical assay data demonstrate that SARS-CoV-2 *Nsp6* can induce glycolysis.

Glycolysis takes place in the cytoplasm and generates energy, in the form of ATP, by converting glucose into pyruvate. Under aerobic conditions, pyruvate can diffuse into the mitochondria, enter the citric acid cycle, and generate additional ATP. Therefore, we next tested if the increased glycolysis observed in flies with heart-specific SARS-CoV-2 *Nsp6* could disrupt the cardiac mitochondria, either indirectly via Nsp6 or directly via the glycolysis pathway enzyme Pgi. We genetically overexpressed either *Nsp6* or *Pgi* (using the heart-specific 4X*Hand* driver) and investigated their impact on mitochondria in the *Drosophila* heart. We found both *Nsp6* and *Pgi* overexpression

caused disorganization of the cardiac actin filaments, and significant loss of mitochondria activity as visualized by ATP5a (Fig. 3d, e). These results indicate that increased glycolysis pathway activity can lead to mitochondrial defects causing heart damage.

Finally, we assessed if increased glycolysis by itself could induce a cardiac phenotype like that caused by heart-specific expression of SARS-CoV-2 *Nsp6* in fly. We genetically overexpressed *Pgi* (heart-specific using the 4X*Hand* driver) to investigate the effect on *Drosophila* heart morphology and function. We found *Pgi* overexpression induced a high mortality rate (29.5%) compared to control flies. In the 4X*Hand*-driven *Pgi* flies that did emerge as adults, we found *Pgi* caused disorganized cardiac actin filaments and reduced cardiac muscle fiber density (Fig. 3d); and a significant loss of mitochondria activity (Fig. 3d, e). In addition, OCT imaging revealed heart-specific *Pgi* overexpression led to reduced heart diastolic diameter and a lengthened heart period, i.e., reduced heart rate (Fig. 3f–h). These results indicate that

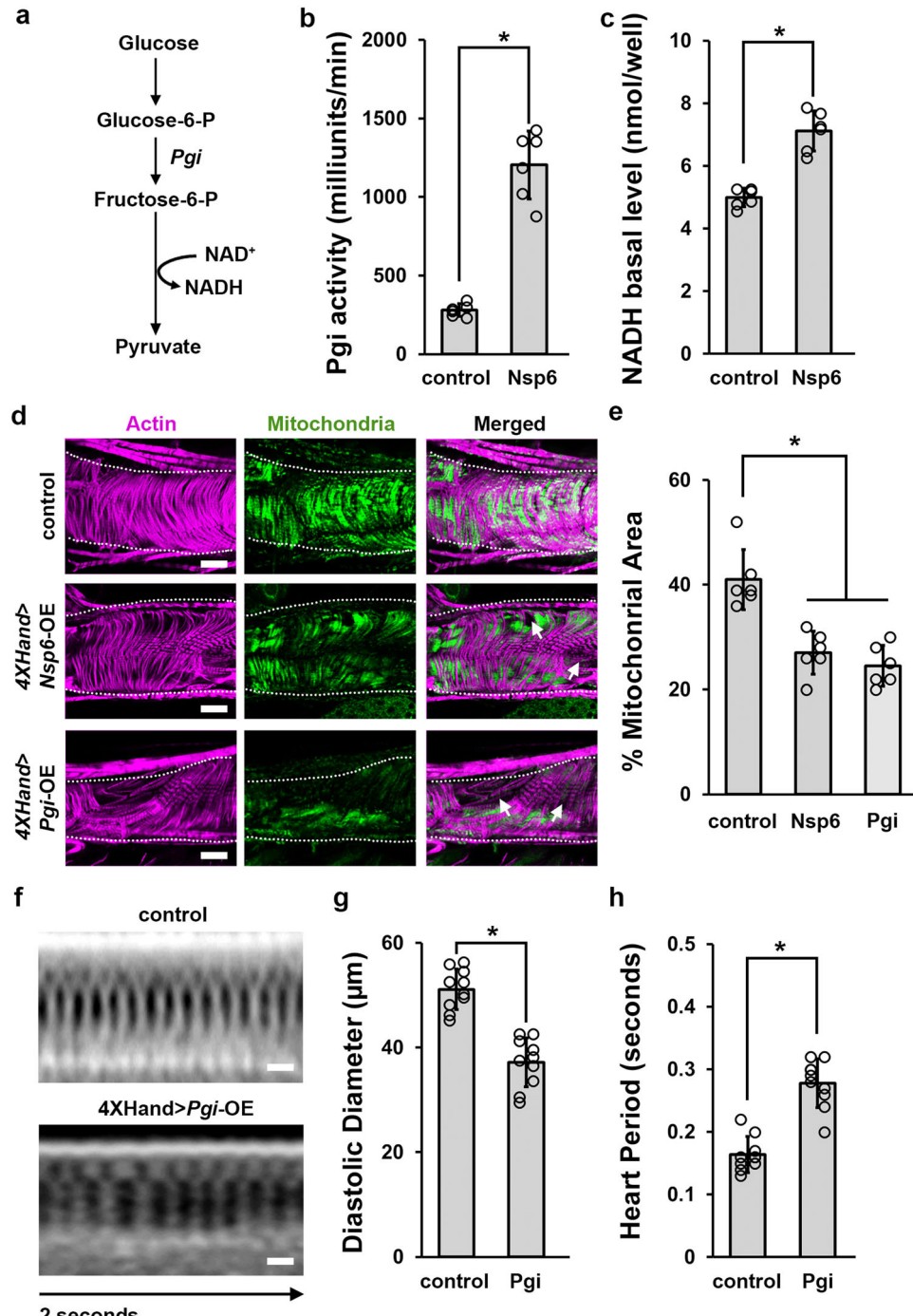

**Fig. 3 SARS-CoV-2 Nsp6 increases glycolysis activity and heart-specific Pgi overexpression causes heart morphological and functional defects. a** Key metabolites in the glycolysis pathway. **b** Quantitation of the enzymatic activity of Phosphoglucose isomerase (Pgi) in flies with ubiquitous (*Tub*-Gal4 driver) overexpression of UAS-*SARS-CoV-2 Nsp6* transgene or wild-type control (*w*[1118]). *P* value = 1.3E−06. *n* = 6 flies per genotype. **c** Quantitation of the NADH level in flies with ubiquitous (*Tub*-Gal4 driver) overexpression of UAS-*SARS-CoV-2 Nsp6* transgene or wild-type control (*w*[1118]). *P* value = 2.4E−05. *n* = 6 flies per genotype. **d** Adult heart mitochondrial phenotype induced by 4X*Hand*-Gal4, cardioblast-specific overexpression (OE) of the UAS-*SARS-CoV-2 Nsp6* or *Pgi* transgenes. Cardiac actin myofibers were visualized by phalloidin staining. Mitochondria were visualized by ATP5a antibody staining. Dotted lines delineate the outline of the heart tube. Arrow points to missing mitochondria. *w*[1118] is a wild-type control. Scale bars (all) = 40 μm. **e** Quantitation of percentage of mitochondrial area in fly heart (see images in **b**). *P* value (control-Nsp6) = 0.01; *P* value (control-Pgi) = 0.001. *n* = 6 flies per genotype. **f** *Drosophila* heartbeat video images from flies induced by cardioblast-specific (4X*Hand*-Gal4 driver) overexpression (OE) of UAS-*Pgi* transgene. *w*[1118] is a wild-type control. Scale bars (both) = 25 μm. **g** Quantitation of adult heart diastolic diameter (in **d**). *n* = 10 flies per genotype. **h** Quantitation of heart period (in **d**). *P* value = 3.5E−07. *n* = 10 flies per genotype. Statistical significance (*) defined as *P* < 0.05 (Student's *t* test in **a**, **e**, **f**; Kruskal–Wallis H-test followed by Dunn's test in **c**); data shown as mean ± SD.

increased glycolysis pathway activity can lead to heart damage, both structurally and functionally.

**SARS-CoV-2 Nsp6 interacts with MGA and the non-canonical polycomb repressive complex.** To identify how glycolysis genes are upregulated and cause the physiological differences upon SARS-CoV-2 *Nsp6* expression, we performed mass spectrometry analysis using human HEK 293T cells that ectopically express mCherry and FLAG-tagged Nsp6. A total of 661 proteins were uniquely detected in the Nsp6 expressing cells, i.e., proteins without any detectable pulldown in the control cells (Supplementary Data 2). Next, to uncover which transcription factors might regulate the underlying transcriptomic changes, and thus are likely the intermediates affected by SARS-CoV-2 Nsp6, we compared the 661 Nsp6-binding proteins against a list of 1639 human transcription factors[42]. This identified nine transcription factors unique to Nsp6 expressing cells (Fig. 4a), suggesting SARS-CoV-2 Nsp6 interacts with these transcription factors to instigate transcriptomic changes that favor the virus.

Among the nine transcription factors, three (MGA, PCGF6 and TFDP1) are part of the non-canonical PRC1.6 complex[43] (Fig. 4b). PRCs are polycomb repressive complexes that play major roles in gene regulation, differentiation, cell cycle control, and development and are known for their varied subunit composition. PRC1.6 specifically is made up of PCGF6 (Polycomb group RING finger protein 6), MGA (MAX gene-associated protein) and E2F6 transcription factors, and L3MBTL2 (a histone-binding protein)[43]. PCGF6 is the defining subunit of this particular PRC, while MGA was shown essential for targeting specific genomic sites as a sequence-specific DNA-binding factor and through its scaffolding function[43]. E2F6 (Transcription factor E2F6) and TFDP1 (Transcription factor Dp-1) form a dimer involved in gene repression[44]. MGA regulates MYC-MAX target gene expression by suppressing transcriptional activation by MYC[44]. In fact, both MGA and MYC partner with MAX and both recognize the same sequence (E-box DNA-motif) resulting in antagonistic action, i.e., the genes targeted, bound and activated by MYC, are bound and repressed by MGA (Fig. 4b, c). Notably, MGA/PCGF6/TFDP1/MAX:MYC/MAX regulated genes include those important for the $G_2$-M checkpoint in the cell cycle and for glycolysis[44]. Based on these previous findings, we hypothesized that SARS-CoV-2 Nsp6 blocks the PRC1.6 complex (which comprises MGA/MAX) through interaction with multiple complex proteins (MGA, PCGF6 and TFDP1) thus releasing its antagonistic effect on MYC/MAX, thereby inducing translation of MYC-pathway target genes which also encode proteins important for glycolysis. To confirm the link between increased MYC and increased glycolysis activity, we ubiquitously overexpressed *Myc*, the fly homolog of human *MYC*, and assayed glycolysis pathway components. *Myc* overexpression significantly increased the activity of Pgi (Fig. 4d) as well as the level of NADH (Fig. 4e) in flies. These findings are similar to the SARS-CoV-2 *Nsp6*-induced dysregulation of glycolysis and suggest that disruption of the MGA/MAX:MYC/MAX balance might act as the Nsp6 mediator in this process.

**Inhibition of glycolysis pathway activity by 2DG attenuates SARS-CoV-2 *Nsp6*-induced heart morphological and functional defects in fly.** Next, we investigated means of pharmacological intervention of the *Nsp6*-induced glycolysis increase and related phenotypes. The compound 2-deoxy-D-glucose (2DG) is a known inhibitor of glycolysis pathway activity by targeting hexokinase and Pgi[45–47] (Fig. 5a). Therefore, we tested whether treating the flies with 2DG could attenuate the Nsp6-induced cardiac defects. Starting at the 1st instar larval stage, the flies were

administered different doses of 2DG through their food. A dose-response curve (0, 2, 10, 30, 50, 75, 100 mM) in control flies showed severe lethality at 50 mM and (near) complete lethality at 75 and 100 mM doses of 2DG (Supplementary Fig. 3). Thus, informing the maximum applicable dose in flies. We tested a range of 2DG doses for their efficacy in the *Nsp6*-transgenic flies (0, 2, 10, 50 mM). Likewise, the 50 mM dose of 2DG was toxic to the *Nsp6*-transgenic flies, resulting in complete lethality (Fig. 5b). The 2 mM dose of 2DG had no detectable effect in either control or *Nsp6*-transgenic flies. However, the 10 mM dose significantly attenuated Nsp6-induced mortality (Fig. 5b) as well as heart morphological (Fig. 5c, d) and functional (Fig. 5e, f) defects. This dose caused a slight, yet significant, increase in mortality in the control flies, but had no significant effect on heart morphology or function in these flies (Fig. 5c–f). Taken together, these results further strengthen the link between SARS-CoV-2 Nsp6 protein, MYC-pathway dysregulation, increased glycolysis, and ultimately cardiac damage.

**Inhibition of glycolysis pathway activity by 2DG attenuates SARS-CoV-2 *Nsp6*-induced functional defect in mouse cardiomyocytes.** Finally, we determined whether our findings in fly translated to a mammalian model. We isolated primary cardiomyocytes from E12.5 mice embryonic heart and transfected the cell cultures with the SARS-CoV-2 *Nsp6* transgene (Fig. 6a). Mice in which glucose metabolism has been disrupted frequently show hypertrophy within their cardiac phenotype[48], therefore we checked the cardiomyocytes for hypertrophy. SARS-CoV-2 *Nsp6* (*Nsp6*-OE) significantly increased *Atrial natriuretic peptide* (*ANP*) and *type B natriuretic peptide* (*BNP*)—both established markers of cardiac hypertrophy—compared to primary cardiomyocytes from the control group (Lipofectamine; Lipo) (Fig. 6b, c). Moreover, the relative expression level of genes involved in glycolysis (*Gpi1*, *Pfkm*, *Pgam2*, and *Eno1*; Fig. 6d) were significantly increased after *Nsp6*-OE (Fig. 6e–h); and like in fly, these could be restored by 2DG supplementation (Fig. 6e–h). Primary cardiomyocytes with SARS-CoV-2 *Nsp6*-OE exhibited significantly increased $Ca^{2+}$ absorption intervals when compared to the control cultures (Lipo) (Fig. 6i, j). $Ca^{2+}$ waves are critical to the muscular contraction and relaxation cycles in the heart and thus for its function[49]. Supplementation with 2DG restored the $Ca^{2+}$ absorption by significantly lowering the intervals in *Nsp6*-OE primary cardiomyocytes (Fig. 6i, j). Together, these results confirmed that SARS-CoV-2 *Nsp6* significantly affects mouse heart function by regulating the glycolysis pathway, as inhibition of glycolysis pathway activity with 2DG significantly attenuated the cardiomyocyte phenotype.

**Discussion**

Cardiovascular complications are widely observed in COVID-19. In fact, increased levels of biomarkers for cardiac injury and abnormal echocardiograms (heart rhythmic function) frequently occur in COVID-19 patients[50,51]—even prior to cardiac symptomatology and in those infected but otherwise asymptomatic[12,13]. The metabolic pathways that regulate heart development and function, including glycolysis, are highly conserved from fly to human[52]. As such, the fly has proven to be a valuable model to study heart failure in humans. Here, we presented evidence that the cardiac phenotypes in COVID-19 might be instigated by SARS-CoV-2 protein Nsp6. Altogether, expression of *Nsp6* increased glycolysis, which led to cardiac dysfunction in both *Drosophila* and mouse models.

Our proteomics data revealed that SARS-CoV-2 Nsp6 directly interacts with host factors that regulate glycolysis, including the MGA, PCGF6 and TFDP1 transcription factors, which are

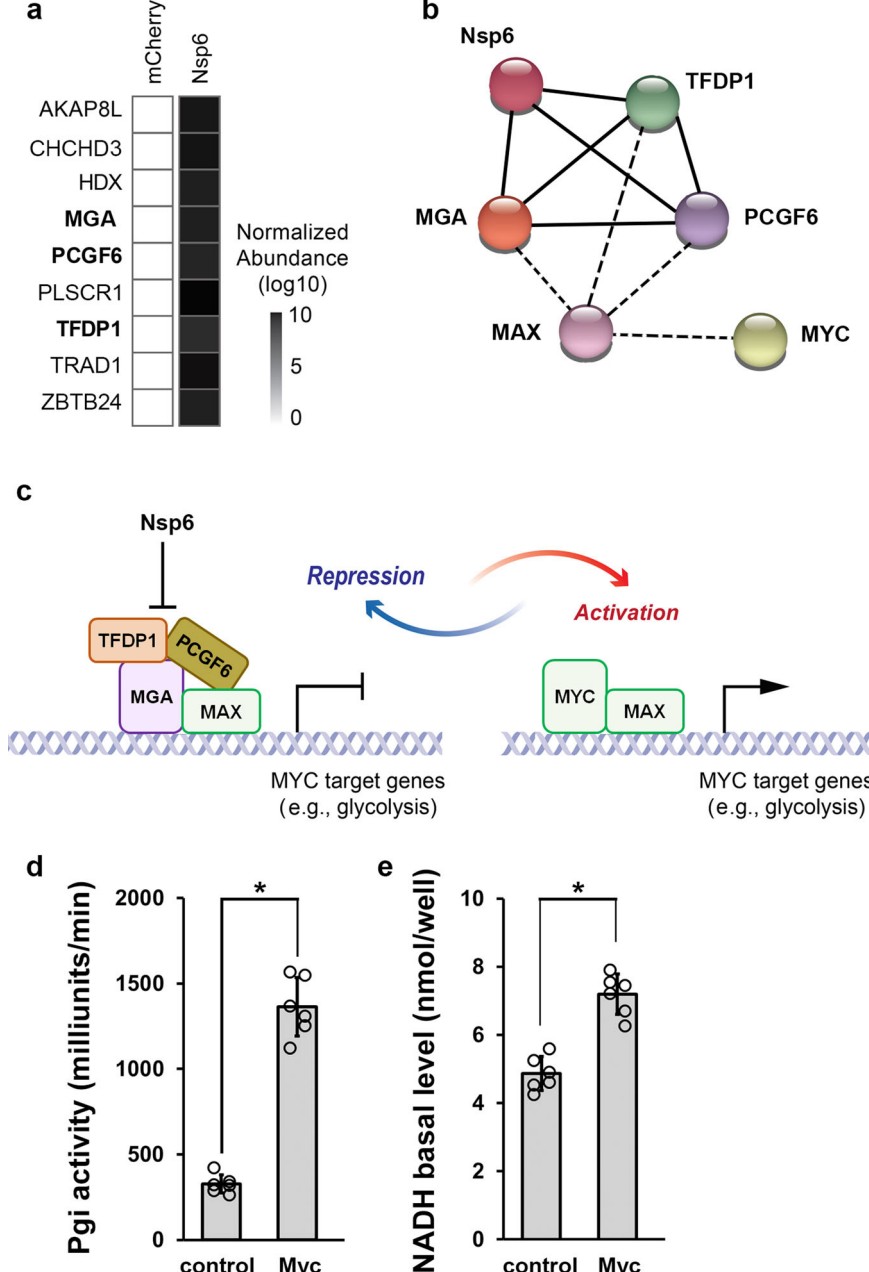

**Fig. 4 SARS-CoV-2 Nsp6 increases glycolysis gene expression by inhibiting an MGA-containing repressive complex. a** Heatmap summarizes the abundance of transcription factors (obtained from mass spectrometry analysis) in the mCherry (left) and SARS-CoV-2 *Nsp6* (right) expressing HEK 293T cells. **b** Known protein-protein interactions among SARS-CoV-2 Nsp6-binding transcription factors, as well as MYC and MAX. Solid line: Interaction identified in this study. Dotted line: Other known interactions from the STRING database. **c** Schematic illustration depicting regulation of glycolysis gene expression by MGA or MYC-containing protein complexes. Quantitation of the enzymatic activity of Phosphoglucose isomerase (Pgi) (**d**; *P* value = 6.4E−08) and NADH (**e**; *P* value = 2.4E−05) levels in flies with ubiquitous (*Tub*-Gal4 driver) expression of the UAS-*Myc* transgene or wild-type control (*w*[1118]). *n* = 6 flies per genotype. Statistical significance (*) defined as *P* < 0.05 (Student's *t* test); data shown as mean ± SD. MGA MAX gene-associated protein, MAX MYC-associated factor X, MYC MYC proto-oncogene, basic helix-loop-helix (BHLH) transcription factor, PCGF6 Polycomb group ring finger 6, TFDP1 Transcription factor Dp-1.

members (or interactors) of the non-canonical PRC1.6 complex[43,44]. MYC-associated factor X (MAX) is another member of this complex. This MGA/PCGF6/TFDP1/MAX (MGA/MAX) complex recognizes and binds the same sequences as those targeted by MYC/MAX, and thus acts as an antagonist of MYC/MAX complex-mediated gene expression. MYC/MAX regulates genes encoding glycolysis pathways proteins, thus inducing glycolysis activity through increased gene expression. By binding the MGA, PCGF6 and TFDP1 transcription factors,

SARS-CoV-2 Nsp6 inhibits formation of the MGA/MAX complex, which shifts the balance towards MYC/MAX (Fig. 7). Our data show that increased MYC can be directly responsible for increased glycolysis, i.e., Pgi activity and NADH basal levels, in fly hearts.

Consistent with more active glycolysis, other genes upregulated by SARS-CoV-2 *Nsp-6* in the fly heart include those involved in the mitochondrial TCA cycle and oxidative phosphorylation (OxPhos). OxPhos generates reactive oxygen species (ROS),

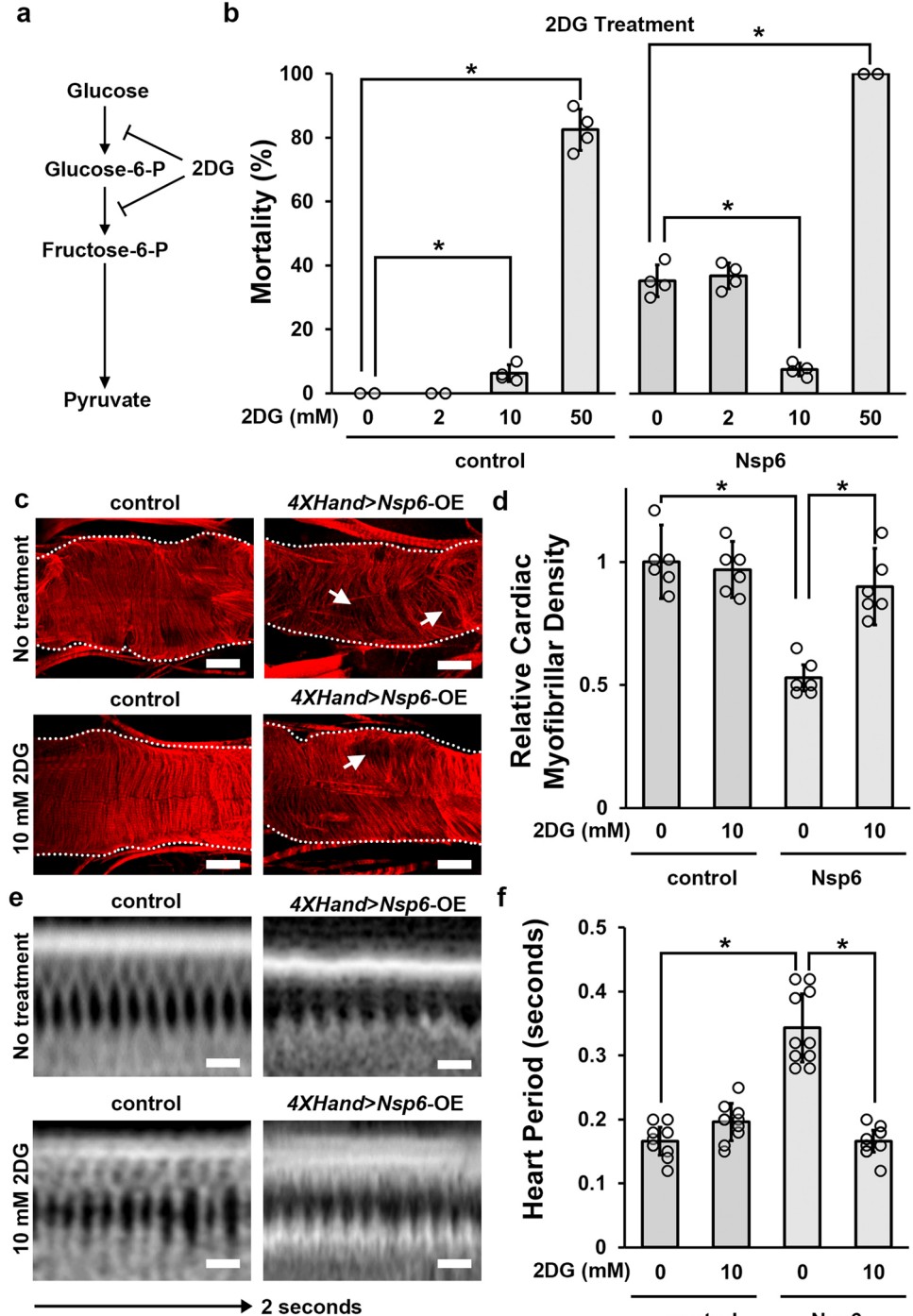

**Fig. 5 Inhibiting glycolysis activity by 2DG attenuates SARS-CoV-2 *Nsp6*-induced heart morphological and functional defects. a** Key metabolites in the glycolysis pathway, with the conversion steps inhibited by 2-deoxy-D-glucose (2DG) indicated. **b** Quantitation of mortality rate prior to eclosion induced by cardioblast-specific (*4XHand*-Gal4 driver) overexpression of the UAS-*SARS-CoV-2 Nsp6* transgene, or wild-type (*w[1118]*) control, following different doses of 2DG. Mortality was calculated as (CyO wing − straight wing)/CyO wing × 100. *P* value (control 0–10 mM) = 0.05; (control 0–50 mM) = 0.002; (Nsp6 0–10 mM) = 0.05; (Nsp6 0–50 mM) = 0.0003. *n* = 4 repeats (~50 flies/vial). **c** Adult heart phenotype induced by cardioblast-specific (*4XHand*-Gal4 driver) overexpression (OE) of the UAS-*SARS-CoV-2 Nsp6* transgene, compared to wild-type (*w[1118]*) control, following different doses of 2DG. Cardiac actin myofibers were visualized by phalloidin staining. Dotted lines delineate the outline of the heart tube. Arrow points to missing cardiac myofibers. Scale bars (all) = 40 µm. **d** Quantitation of adult heart cardiac myofibrillar density (in **b**) relative to control. *P* value (0 mM control-Nsp6) = 0.0006; (Nsp6 0–10 mM) = 0.03. *n* = 6 flies per genotype. **e** *Drosophila* heartbeat video images from flies carrying cardioblast-specific (*4XHand*-Gal4 driver) overexpression (OE) of the UAS-*SARS-CoV-2 Nsp6* transgene, or wild-type (*w[1118]*) control flies, following different doses of 2DG. Scale bars (all) = 25 µm. **f** Quantitation of heart period (in **d**). *P* value (0 mM control-Nsp6) = 0.00001; (Nsp6 0–10 mM) = 0.00001. *n* = 10 flies per genotype. Statistical significance (*) defined as *P* < 0.05 (Kruskal–Wallis H-test followed by Dunn's test); data shown as mean ± SD.

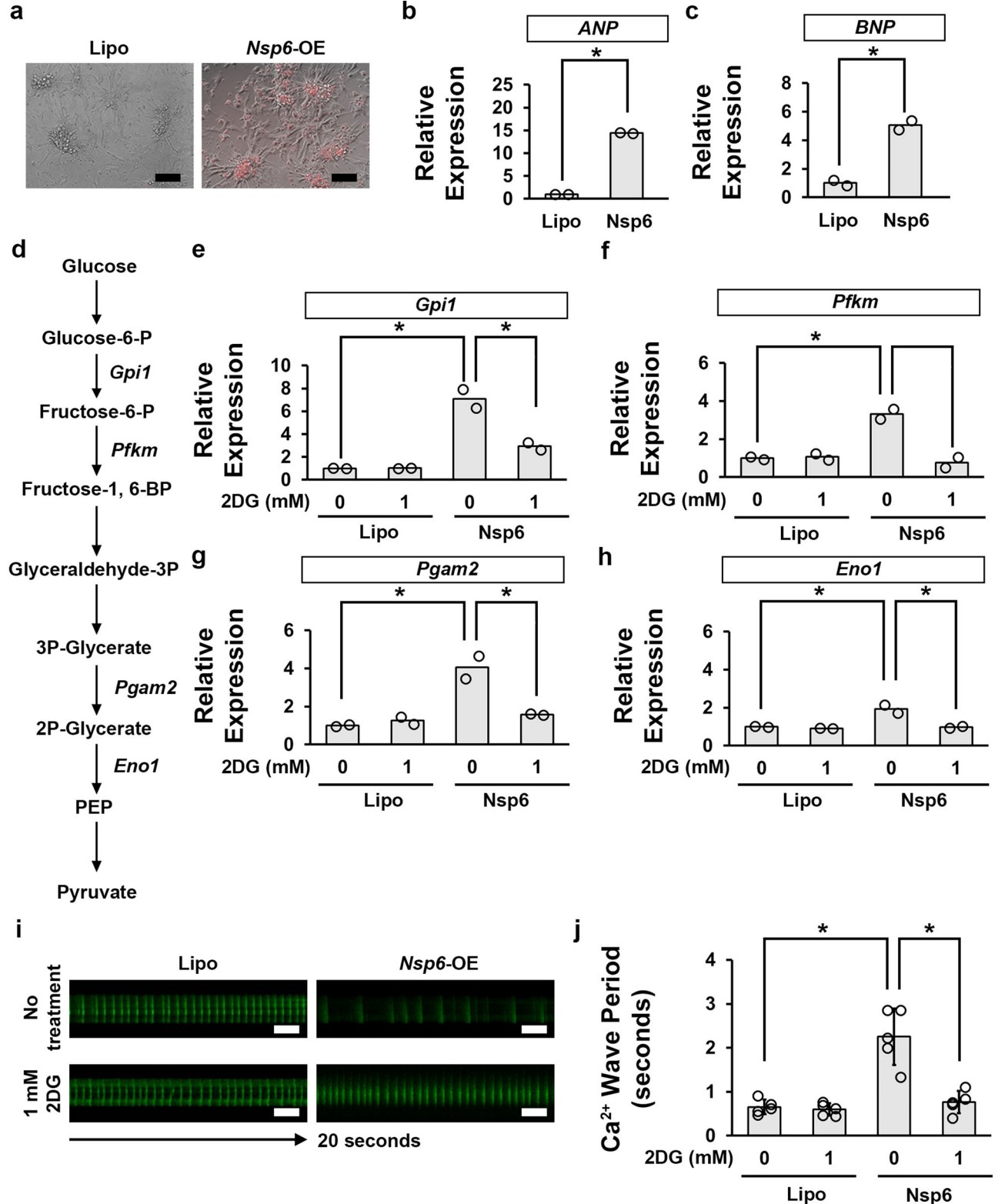

which have been associated with several physiological functions as well as pathology—like the sterile inflammatory response often observed in heart failure[53]. Two studies have demonstrated a role for the HIF1α-glycolysis axis in SARS-CoV-2 infection, but neither has specified the viral protein responsible. The first study, showed SARS-CoV-2 induces mitochondrial ROS in monocytes, during later stages in the viral life cycle[54]. The second study found increased activity of the HIF1α-glycolysis fatty acid synthesis axis

in an in vitro human airway epithelium model; they used ciliated-like cells in human pluripotent stem cell-derived airway organoids, which were permissive to SARS-CoV-2 infection[55]. Our findings show dysregulated glycolysis in the heart related to SARS-CoV-2; specifically, we demonstrate that the viral Nsp6 protein is responsible, suggesting this might be the culprit in lung epithelium as well. In fact, we found *sima*, the *Drosophila* ortholog of *HIF-1a*, was significantly upregulated in fly hearts

**Fig. 6 Inhibiting glycolysis activity by 2DG attenuates SARS-CoV-2 _Nsp6_-induced heart hypertrophy and functional defects in mouse primary cardiomyocytes. a** Mouse primary cardiomyocytes were transfected with the _SARS-CoV-2 Nsp6_ transgene. The _SARS-CoV-2 Nsp6_ transgene was visualized in red. Lipofectamine only treated mouse primary cardiomyocytes (Lipo) served as the control. Scale bars (both) = 100 μm. **b** Quantitation of hypertrophy marker _ANP_ relative to control. $P$ value = 3.7E−05. ANP, Atrial natriuretic peptide. $n$ = 2 mice (cardiomyocytes) per condition. **c** Quantitation of hypertrophy marker _BNP_ relative to control. $P$ value = 0.008. BNP, type B natriuretic peptide. $n$ = 2 mice (cardiomyocytes) per condition. **d** Key metabolites in the mammalian glycolysis pathway. **e–h** Graphs Quantitation of the glycolysis gene expression levels in mouse primary cardiomyocytes transfected with the _SARS-CoV-2 Nsp6_ transgene, or lipofectamine (Lipo) control mouse primary cardiomyocytes. _Gpi1_ (**e**) $P$ value (0 mM Lipo-Nsp6) = 0.03; (Nsp6 0–1 mM) = 0.04. _Pfkm_ (**f**) $P$ value (0 mM Lipo-Nsp6) = 0.01; (Nsp6 0–1 mM) = 0.02. _Pgam2_ (**g**) $P$ value (0 mM Lipo-Nsp6) = 0.04; (Nsp6 0–1 mM) = 0.05. _Eno1_ (**h**) $P$ value (0 mM Lipo-Nsp6) = 0.05; (Nsp6 0–1 mM) = 0.05. $n$ = 2 mice (cardiomyocytes) per condition. **i** Mouse primary cardiomyocyte $Ca^{2+}$ wave video images from cells transfected with the _UAS-SARS-CoV-2 Nsp6_ transgene, or lipofectamine (Lipo) control mouse primary cardiomyocytes. Scale bars (all) = 200 μm. **j** Quantitation of the $Ca^{2+}$ wave rate (in **d**). $P$ value (0 mM Lipo-Nsp6) = 0.006; (Nsp6 0–1 mM) = 0.03. $n$ = 5 primary cells (cardiomyocytes) per condition. Statistical significance (*) defined as $P < 0.05$ (Student's $t$ test in **b**, **c**; Kruskal–Wallis H-test followed by Dunn's test in **e–h**, **j**); data shown as mean ± SD for $n > 5$.

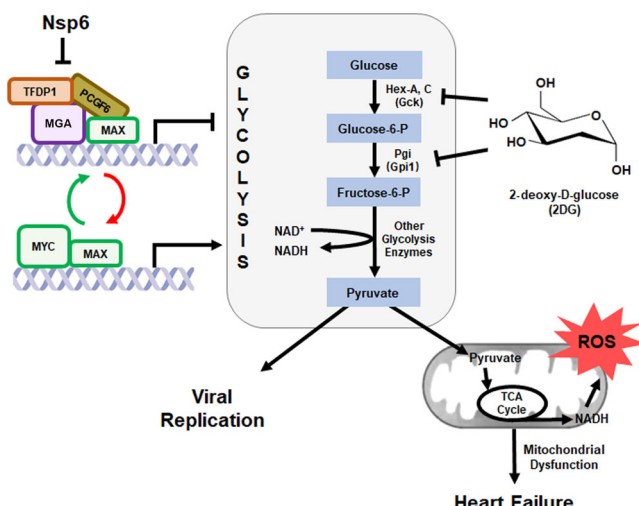

**Fig. 7 Model of SARS-CoV-2 Nsp6-mediated disruption of glycolysis.**
Graphic depiction of proposed model by which SARS-CoV-2 Nsp6 disrupts the MGA/MAX:MYC/MAX balance thereby increasing glycolysis pathway activity, which in turn leads to heart failure and potentially facilitates virus replication. The model also indicates where 2-deoxy-D-glucose (2DG) might intervene and inhibit the Nsp6-glycolysis-induced cardiac phenotype. MGA MAX gene-associated protein, MAX MYC-associated factor X, MYC MYC proto-oncogene, basic helix-loop-helix (BHLH) transcription factor, PCGF6 Polycomb group ring finger 6, ROS reactive oxygen species, TCA cycle tricarboxylic acid cycle (a.k.a. Krebs cycle), TFDP1 Transcription factor Dp-1, Hex-A/Hex-C Hexokinase-A/-C, Phosphoglucose isomerase (Pgi).

expressing SARS-CoV-2 Nsp6 (RNAseq, 1.47 fold, adj. $P = 0.0364$). Increased ROS triggers the hypoxia-inducible factor-1a (HIF-1a)-dependent pathway which in turn upregulates glycolysis genes[54],[55]. This could potentially provide a feedback loop reinforcing the initial increase in glycolysis activity by Nsp6. During heart failure, cardiomyocytes switch from fatty acid to glucose metabolism in order to optimize the ATP production to oxygen consumption ratio[48]. Thus, while increased glycolysis activity acts as a protective mechanism under cardiac pathogenic conditions, it might only exacerbate the overactive glycolysis during SARS-CoV-2 infection induced by its Nsp6 protein. Further studies are needed to delineate these observations.

How these findings translate to new SARS-CoV-2 variants remains speculative. It appears that infection with the recent Omicron variant causes less severe disease than prior variants. Although, the contribution of prior infection with SARS-CoV-2 and/or vaccination to this effect remain to be determined. Notably, several mutations in Nsp6 Omicron have been identified

(https://covdb.stanford.edu/variants/omicron/). It is conceivable that these mutations affect Nsp6 binding to the host MGA, PCGF6 and TFDP1 transcription factors and thus its ability to inhibit formation of the MGA/MAX complex and regulate glycolysis. Thereby rendering the Nsp6 Omicron variants less toxic; however, this remains speculative. One of the advantages of the fly system is its ability to rapidly model these new viral variants. Their pathogenicity can then be compared across variants for multiple relevant tissues.

SARS-CoV-2 Nsp6 upregulated glycolysis as well as related pathways, such as TCA and OxPhos in the mitochondria, and the pentose phosphate pathway (PPP) in the cytosol. In addition to generating bioenergy in the form of ATP, these processes generate metabolites to support various biological processes, including biosynthesis and protein modification. For example, glucose-6-P, an intermediate of glycolysis, also provides the starting metabolite for the PPP. The PPP mediates de novo purine synthesis which is thought to play a role in virus replication by providing essential metabolites. In fact, a recent publication showed glucose was depleted in cells (VeroE6) infected with SARS-CoV-2 and demonstrated the virus hijacks the folate and one-carbon metabolism pathway to favor de novo purine synthesis and to support replication early in the viral life cycle[56]. Similarly, our data show that genes belonging to 'purine metabolic processes' and 'one carbon pool by folate' are significantly upregulated in Nsp6 expressing fly hearts. These data imply that increased glycolysis supports SARS-CoV-2 virus replication. Moreover, our findings suggest SARS-CoV-2 Nsp6 protein mediates this process (Fig. 7).

Altered glycolysis following SARS-CoV-2 infection has been reported in human cells and patients, such as in human Caco-2 cells, which are highly permissive to infection[57]. Host-dependent dysregulation of glycolysis, among other metabolic pathways, following SARS-CoV-2 infection was also evident in transcriptomic data obtained from different human respiratory cell lines and various patient samples[58]. In addition, single-nucleus RNAseq data from lung tissue of deceased COVID-19 patients found changes in genes related to several metabolic pathways, including glycolysis and OxPhos[59]. Besides, mild ischemia (early stages) is characterized by increased glycolytic flux with concomitant increased glucose uptake[48]. Furthermore, an in vivo study in hamsters infected with SARS-CoV-2 found increased expression of ROS-related genes in heart tissue[60]. Of note, viral transcripts were detected in the left ventricle and atrium, and the right atrium, but not the right ventricle; this suggests that the sampling site might matter. The same study also found significantly altered ROS in heart samples of deceased patients with COVID-19 compared to samples from individuals without COVID-19[60]. These findings are in line with those we made in fly. In fact, increased mitochondrial dysfunction and increased

ROS are well-known contributors to cardiac disease[61]. Together, the direct (in vivo animal models) and indirect (human in vitro and patient non-heart tissue) data support the notion of SARS-CoV-2 Nsp6-induced glycolysis, and ROS, in the heart during COVID-19 (model in Fig. 7).

The data make a case for SARS-CoV-2 Nsp6-induced dysregulated glycolysis driving cardiac pathology. However, the cardiac injury, and ultimately heart failure, in COVID-19 could be a secondary effect stemming from the increased thrombotic activity or cytokine storm associated cardiomyopathy, rather than a primary effect directly caused by virus infection. Virus in cardiac tissue has been reported in in vivo mammalian models. In (6/9) hamsters intranasally infected with SARS-CoV-2, cardiomyocytes were found damaged, and SARS-CoV-2 was detected within the cells[60]. Mice, unlike hamsters, are not naturally susceptible to SARS-CoV-2 infection. However, transgenic mice that express HFH4-hACE2—the human ACE2 receptor that facilitates virus entry—showed cardiac pathology following SARS-CoV-2 infection, and low viral RNA copies were detected in the hearts of some of the mice[62].

Findings in human patients have repeatedly shown virus present in cardiac tissue; but, virus entry of cardiomyocytes specifically remains a matter of debate. One early study reported no evidence for relevant viral presence (the meaning of "relevant" was not defined) in cardiac tissue in 40 deceased patients diagnosed with severe SARS-CoV-2 infection during the first wave of the pandemic[4]. Yet, findings that support viral entry into cardiac tissue have been steadily accumulating. Early studies detected SARS-CoV-2 genomes in endomyocardial biopsies of patients with cardiac injury[1,3]. In addition, multiple autopsy studies have reported direct evidence of SARS-CoV-2 presence in heart tissue in those infected with the virus [16/39 (41.0%) deceased patients positive for SARS-CoV-2 infection[2]; 8/8 deceased patients with COVID-19 diagnosis and morphological signs of myocarditis[7], including histopathology to show SARS-CoV-2 present in the cardiomyocytes]. A large study reported 49/97 (48%) autopsies with SARS-CoV-2 RNA in the cardiomyocytes[5] and case reports similarly found SARS-CoV-2 present in cardiomyocytes of heart tissue obtained from deceased patients with COVID-19 [six patients[63]; one patient[64,65]]. However, while one of the larger studies did detect relevant SARS-CoV-2 virus load in heart tissue in 41/95 deceased (43%; RNA sequencing), it was co-localized with wheat germ agglutinin (WGA) suggesting the virus was present in interstitial cells surrounding the large cardiomyocytes. Only rarely did they detect viral RNA within the cardiomyocytes[6]. Therefore, while other pathomechanisms, such as hypoxemia caused by respiratory failure, (hyper)inflammation, or macrophages originating in the lungs, undoubtedly contribute to cardiac distress. Together, the findings support direct action of SARS-CoV-2 in cardiac tissue, and within cardiomyocytes, as a cause of the cardiac pathology observed in patients with COVID-19.

Our findings are in line with several human cell culture studies that showed increased glycolysis following infection with SARS-CoV-2. In human Caco-2 cells, SARS-CoV-2 infection affected the carbon metabolism[57]. Treating the cells with 2DG, a potent inhibitor of the glycolysis pathway, prevented virus replication in vitro, and induced changes in an endoplasmic reticulum protein known to regulate lipid metabolism[57]. In human pluripotent stem cell-derived airway organoids, the small molecule GW6271 (FDA-approved) both blocked SARS-CoV-2 infection and virus replication; GW6271 acts by inhibiting the HIF1α-glycolysis axis[55]. A study in Vero E6 cells found that 2DG treatment reduced SARS-CoV-2-induced glycolysis activity, attenuated cytopathy, and reduced viral replication[66]. A recent study in healthy control mice showed that 2DG treatment-induced metabolic changes in the mouse heart would oppose those previously observed in SARS-CoV-2 infected cells[67]; thus, providing indirect support for 2DG in treating SARS-CoV-2-associated cardiac disease. Our findings expand on these previous studies by providing in vivo data (fly; and mouse ex vivo) that demonstrate SARS-CoV-2 Nsp6 protein can cause a cardiac phenotype, this phenotype is marked by increased glycolysis activity—due to interaction of Nsp6 with a host protein complex that regulates glycolysis—and dysfunctional mitochondria, and this cardiac phenotype can be effectively treated with 2DG.

Regards 2DG treatment for COVID-19 in humans, a combination therapy of 2DG and low dose radiation has been proposed to treat the COVID-19-associated cytokine storm[68]. In India, 2DG treatment has received emergency use approval to curb a devastating recent COVID-19 outbreak[69]; a clinical trial in India to investigate the effectivity of 2DG to treat COVID-19 has been registered with WHO (registration: CTRI/2021/01/030231). Moreover, WP1122, a 2DG derivative is pursued as a lead compound against COVID-19 by targeting glycolysis to inhibit virus replication (Moleculin Biotech, TX, USA). One of the main hurdles to start clinical trials is the requirement for supporting data from animal models. The outcomes of this trial will be a major first step to establish the potential of 2DG as a COVID-19 therapeutic.

Of note, another strategy to inhibit glycolysis is to lower the available glucose levels by a ketogenic diet. A study in mice (mCoV-A59 driven infection) showed protection induced by the ketogenic immunometabolic switch[70], and preliminary findings from an early clinical trial showed reduced severity (based on the need for an intensive care unit and death) in patients with COVID-19 on a ketogenic diet compared to those on a eucaloric standard diet[71,72]. These are preliminary results, and mostly focused on immunological measures; however, additional clinical trials are under way (https://clinicaltrials.gov) and will hopefully shed light on the effect of a ketogenic diet on the COVID-19-associated cardiac pathological outcomes.

Taken together, our findings demonstrate virus control of host glycolysis and related pathways in the heart (fly and mouse)—likely to benefit virus replication—that this pathology is mediated by SARS-CoV-2 Nsp6 protein, and that treatment with 2DG provides a promising therapeutic strategy for cardiovascular pathology in COVID-19.

## Methods

**Fly strains**. All Drosophila fly stocks were reared and kept on standard food at 25 °C. 4XHand-Gal4/CyO ['Curly O' (CyO) balancer chromosome with curly wing phenotype] and UAS-SARS-CoV-2 transgenic flies were generated previously[33]. The other fly lines were obtained from the Bloomington Drosophila Stock Center (Indiana University, IN): w[1118] (#BL-3605), UAS-Pgi-OE (#BL-60676), and UAS-Myc-OE (#BL-9674, and #BL-64759).

**Mortality at eclosion**. To assay the effect of viral gene expression on fly viability, a balancer system was used. Drosophila lines that overexpressed (OE) a UAS-SARS-CoV-2 gene were crossed with the 4XHand-Gal4/CyO line. Offspring either carried UAS-SARS-CoV-2 gene-OE/CyO in which the balancer chromosome resulted in a CyO curly wing phenotype without transgene expression, or they carried UAS-SARS-CoV-2 gene-OE/4XHand-Gal4 resulting in transgenic lines with straight wings that express the SARS-CoV-2 gene driven by 4XHand-Gal4 for heart-specific expression. Embryo progenies were collected and allowed to develop under standard conditions. Mortality at eclosion (adult emergence from pupa stage) was based on the percentage of flies with SARS-CoV-2 gene expression (straight wing) that failed to emerge as adults, relative to siblings that did not express the SARS-CoV-2 gene construct (CyO wing). Results are presented as a Mortality Index calculated by: (CyO wing − straight wing)/CyO wing × 100.

**Adult survival assay**. Following egg-laying, Drosophila larvae were kept at 25 °C, standard conditions and an optimal temperature for UAS-transgene expression. Adult male flies were maintained in vials in groups of 20 or fewer. The number of living flies in each group was recorded every second day. The assay was carried out

until no survivors were left for any of the fly lines. Per genotype 100 flies were assayed.

**Drosophila heart imaging**. Adult flies were dissected and fixed for 10 min in 4% paraformaldehyde in phosphate-buffered saline (1× PBS). Flies were incubated in block solution containing 0.1% Triton X-100 plus 2% bovine serum albumin (BSA) in 1× PBS for 40 min, then incubated with either Alexa Fluor 647 phalloidin (Thermo Fisher; 1:1,000 in block solution) by itself or combined with primary antibody ATP5a (Abcam, ab14748; 1:1,000 in block solution) overnight at 4 °C. Next day, the flies were washed three times with 1× PBS, then imaged or for immunostaining incubated with secondary Alexa Fluor 488 goat anti-mouse antibody (Thermo Fisher; 1:1,000 in block solution) for 2 h at room temperature, and then washed three times with 1× PBS. Confocal imaging was performed using a ZEISS LSM 900 with Airyscan 2 and a 63× Plan-Apochromat 1.4 N.A. oil objective (ZEN image acquisition software). For quantitative comparisons of fluorescence intensity, common settings were chosen to avoid oversaturation. ImageJ software (version 1.49) was used to process the images. For quantitative comparisons of cardiac muscle fiber density, we analyzed six control flies and six over-expressing flies of each genotype.

**Optical coherence tomography (OCT) measurements and analysis of cardiac function**. Cardiac function in adult *Drosophila* was measured using an optical coherence tomography (OCT) system (Bioptigen Inc.). For this, 4-day-old flies were anesthetized by carbon dioxide (CO2) for 3–5 min and females were pre-selected from each group. Each fly was gently placed on a plate with petroleum jelly (Vaseline) for immobilization with the dorsal aspect facing the OCT microscopy source, then rested for at least 10 min to ensure the fly was fully awake. For each genotype, 10 control and 10 over-expressing flies were used. OCT was used to record the adult heart rhythm and heart wall movement at the same position, i.e., the cardiac chamber in abdominal segment A2 of each fly. M-mode images recorded the heart wall movement during the cardiac cycle. ImageJ software (version 1.49)[73] was used to process the images. The diastolic dimension and systolic diameter were processed, measured, and determined based on three consecutive heartbeats. The heart rate was determined by counting the total number of beats that occurred during a 15-second recording.

**RNA extraction and sequencing of *Drosophila* heart cells**. For each sample, hearts from fifty-one—week-old adult flies were manually dissected and collected directly into ice-cold TRIzol LS reagent (Thermo Fisher Scientific). Then, RNA extraction was performed according to the instructions provided by the manufacturer (Direct-zol RNA Microprep, Zymo Research). RNA quality and concentration were analyzed by agarose gel electrophoresis, NanoDrop 8000 Spectrophotometer, and Agilent 2100. More than 400 ng of total RNA for each sample was used in subsequent library preparation and sequencing. Carried out in duplicate.

Illumina-based RNAseq library preparation and subsequent paired-end sequencing were performed by Novogene (Beijing, China). Novogene also carried out mapping of the short-reads, their quantification, and initial differential analyses. In brief, reads that contained Illumina-adaptor sequences or those considered to be of poor quality (>10% uncertain base nucleotides, or Phred score <5 for over 50% of the reads) were removed before mapping. The short-reads, generated from the RNAseq, were mapped to the Berkeley Drosophila Genome Project (BDGP) reference genome release 6 (BDGP 6) using HISAT2 version 2.0.5[74] using parameters: --dta --phred33. For gene-level quantification, the *Drosophila* gene annotation model from Ensembl version 100[75], which corresponds to Flybase 6.28[76], was used with featureCounts version 1.5.0-p3 (with default parameters). DESeq2 version 1.2.0 was used for differential expression analysis[77].

ClusterProfiler version 3.18.1[78] was used for the Gene Ontology (GO) and KEGG pathway analyses. Classification of glycolysis, pentose phosphate pathway (PPP), citric acid (a.k.a. TCA) cycle, and oxidative phosphorylation (OxPhos) were based on KEGG database annotations[79]. The gene conservation scores (fly-human) were obtained from the DRSC Integrative Ortholog Prediction Tool (DIOPT) version 8[80].

**Pgi activity and NADH level measurement**. The enzymatic activity of Phosphoglucose isomerase (Pgi) and NADH levels were determined using the Phosphoglucose Isomerase Colorimetric Assay Kit (Sigma) according to the manufacturer's protocol. For each sample, one fly was homogenized and diluted 1:50. Six samples were measured for each genotype. The colorimetric reaction was measured at 450 nm on a Spark multimode microplate reader (Tecan, Switzerland; SparkControl software, v2.3).

**Mass spectrometry in HEK 293T cells**. Cell culture, sample preparation, mass spectrometry and data analysis were carried out using established protocols[81]. HEK 293T cells (human embryonic kidney-derived 293T cell line; ATCC) were cultured and maintained in DMEM medium (Corning cellgro) containing 10% fetal bovine serum and Penicillin-Streptomycin (10 units/ml). The SARS-CoV-2 *Nsp6* cDNA fragment was codon-optimized and assembled into the pCMV6 vector for C-terminal tagging with mCherry (mCh)-FLAG. The FLAG-tag was used for

affinity purification using FLAG agarose beads (Sigma). The proteins thus pulled down from 1×(10^7) HEK 293T cells were solubilized in 5% sodium deoxycholate, then washed, reduced, alkylated, and trypsinized on the filter[82,83]. Tryptic peptides were separated on a nanoACQUITY UPLC analytical column (BEH130 C18, Waters) over a 165 min linear acetonitrile gradient (3–40%) with 0.1% formic acid on a Waters nano-ACQUITY UPLC system and analyzed on a coupled Thermo Scientific Orbitrap Fusion Lumos Tribrid mass spectrometer[84]. Label-free quantification was performed using Minora, an aligned AMRT (Accurate Mass and Retention Time) cluster quantification algorithm (Thermo Scientific, 2017) using extracted ion chromatograms. Protein abundance was measured by Hi3 method[85]. Quantitation between samples was normalized by mCh abundance.

Based on a previous study[42] we identified the transcription factors present among the human proteins that bind SARS-CoV-2 Nsp6 in our mass spec results. Physical interactions among these transcription factors, as well as MYC and MAX, were determined and visualized using the STRING database version 11[86].

**Treatment with 2-deoxy-D-glucose**. Compound 2-deoxy-D-glucose (2DG) (SIGMA, SIG-D8375) was dissolved in water and added to standard fly food at various concentrations (2, 10, 30, 50, 75 or 100 mM 2DG dilution). For controls, water alone (0 mM) was added to the fly food. Flies were treated from the 1st instar larval stage.

**Mice**. The procedures for animal use were in compliance with all relevant ethical regulations and approved by the University of Maryland School of Medicine Institutional Animal Care and Use Committee (IACUC). Wild-type C57BL/6J mice were purchased from the Jackson Laboratory (Bar Harbor, ME). Male and female mice were paired and observed for pregnancy by the presence of the vaginal plug (E0.5). At E12.5, pregnant female mice were euthanized, and embryos were collected for primary cardiomyocyte isolation.

**Primary cardiomyocyte isolation and culture**. Mice embryonic primary cardiomyocytes were isolated using the Pierce Primary Cardiomyocyte Isolation Kit (Thermo Fisher Scientific). At E12.5 littermates were sacrificed to collect embryonic hearts ($n = 8$). Briefly, E12.5 mouse hearts were collected individually and washed three times with 1× PBS and Hank's Balanced Salt Solution (HBSS). Washed hearts were treated with lysis buffer and incubated at 37 °C with 5% $CO_2$ for 30 min. Lysis buffer was removed and the cells were washed with ice-cold HBSS, twice. Then, prewarmed cardiomyocyte culture medium was added to each heart cell lysate and suspended by gently breaking up the cells with a pipette. Finally, the mouse primary cardiomyocytes were seeded in a Matrigel-coated 6-well culture plate at $1 \times 10^6$ cells/well density and cultured at 37 °C with 5% $CO_2$. After 48 h in culture, the cardiomyocytes were transfected (using Lipofectamine 2000 transfection reagent; Invitrogen) with either control pcDNA3 plasmid (Lipo; control) or SARS-CoV-2 *Nsp6* plasmid, and then incubated overnight. The next day, culture medium was replaced with fresh medium and the cells were incubated for 48 h with (1 mM) or without 2DG treatment. Following, the cells were collected for further morphological and molecular analysis (cardiomyocytes from $n = 2$ mice per condition; control ±2DG, Lipo ±2DG).

**RNA extraction, reverse transcription and RT-PCR**. Mouse primary cardiomyocyte RNA was extracted using the RNeasy Mini Kit (Qiagen) and reverse transcribed using q-script cDNA Supermix (QuantaBio Biosciences) following the manufacturer's guidelines and protocol. RNA samples were analyzed for quality and concentration using a NanoDrop ND-1000 Spectrophotometer. Next, expression of the following genes was analyzed in mice E12.5 primary cardiomyocytes: *ANP*, *BNP*, *Gpi1*, *Pfkm*, *Pgam2*, and *Eno1* with GAPDH as an endogenous reference gene (Supplementary Table 1). The lipofectamine treated cells served as control to the cardiomyocytes transfected with SARS-CoV-2 *Nsp6*. The RT-PCR was performed on Step One Plus System (Applied Biosystem, Grand Island, NY), using the SsoFast EvaGreen Supermix (Bio-Rad) with a total of 20 µl reaction volume: 2 µl cDNA, 4 µl SsoFast EvaGreen Supermix, 2 µl F/R primer mix, and 12 µl water, followed by a standard touchdown PCR protocol.

**Primary cardiomyocyte contraction and Ca²⁺ wave analyses**. The previous prepared primary cardiomyocytes were loaded with 5 µM fluo 4-AM (Thermo Fisher Scientific) for 20 minutes at room temperature and replaced with normal Tyrode's solution containing 140 mM NaCl, 10 mM HEPES, 0.5 mM MgCl2, 0.33 mM NaHPO4, 5.5 mM glucose, 1.8 mM CaCl2, and 5 mM KCl (pH 7.4). The contraction rate (beating) and $Ca^{2+}$ absorption waves of the mouse primary cardiomyocytes were analyzed through confocal microscope using a ZEISS LSM 900 with Airyscan 2 and a 63× Plan-Apochromat 1.4 N.A. oil objective (ZEN image acquisition software). Contractions of the cardiomyocytes were manually counted for the number of beats in 15 seconds.

**Statistics and reproducibility**. Statistical analysis was performed using PAST software (Natural History Museum, Norway). Mean values are presented along with their standard deviation (SD). The Student's *t* test was used for comparisons between two groups. The Kruskal–Wallis H-test followed by a Dunn's test was used

for comparisons between multiple groups. Statistical significance was defined as $P < 0.05$. Details for sample size and replicates used for quantitation and the statistical tests applied to determine significance have been provided in the figure legends.

**Reporting summary.** Further information on research design is available in the Nature Research Reporting Summary linked to this article.

## Data availability

The datasets generated and analyzed during this study have been deposited at public servers. The RNAseq data can be accessed through the NCBI Gene Expression Omnibus (GEO) with the following accession number: GSE17835[87]. The mass spectrometry proteomics data have been deposited to the ProteomeXchange Consortium via the PRIDE[88] partner repository with the dataset identifier PXD036447[89]. The raw data used to generate the graphs in this manuscript have been included as Supplementary Data 3. Any additional data and materials generated during this study are available upon reasonable request.

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

## Acknowledgements
We thank Dr. Na-Young Rho for providing the cell cultures used for $Ca^{2+}$ imaging. We thank the Bloomington Drosophila Stock Center at Indiana University Bloomington (IN) for the *Drosophila* lines. This work was supported, in part, by the University of Maryland Baltimore Institute for Clinical and Translational Research (UMB ICTR) COVID-19 Accelerated Translational Incubator Pilot (ATIP) grant to Z.H., and by the University of Maryland School of Pharmacy Mass Spectrometry Center (SOP1841-IQB2014).

## Author contributions
Z.H. and J.Z. designed the study; J.Z., G.W., X.H., H.L., W.H., Penghua Y. and J.L. carried out the experiments; J.Z., G.W., X.H., H.L., J.L., J.vdL., M.A.K., Peixin Y. and Z.H. analyzed and interpreted the data; J.Z., X.H., H.L. and J.L. prepared the figures; J.vdL. and Z.H. drafted and edited the manuscript. All authors read and approved the final version of this manuscript for publication.

## Competing interests
The authors declare no competing interests.
