## [Peer Review File · Communications Biology]

Reviewers' comments:

Reviewer #1 (Remarks to the Author):

The presented manuscript investigates the mechanisms of SARS-CoV-2 infection and describes possible therapeutic treatments. The authors build their conclusions on the results obtained from various model systems and research areas, demonstrating their broad scientific approach. Using *Drosophila* to study the detrimental effect of SARS-CoV-2 proteins represents a very efficient way to model the host-virus interactions. Overall, the manuscript represents a very detailed study of COVID19 disease consequences. The data and the discussion reflect the complexity of various symptoms and offer a strong hypothesis of the mechanism of cardiovascular complications. The manuscript can be accepted for publication after the authors address the following comments:

1. The authors use UAS/GAL4 system to express the viral genes specifically in the heart tissue. This is a legitimate way to introduce and study the effects of viral proteins on the tissue of interest. Hand-GAL4 drives the viral gene expression starting from the early stages of the development to adulthood, therefore the heart tissue develops with the viral protein being present. However, viral infection occurs when the tissue is completely formed. This difference between real viral infection and the presented model of viral infection should be acknowledged in the introduction, or at the beginning of the Results section.

2. Lower panel Fig 1D does not have a scale bar.

3. Line 201 section, Glycolysis genes are upregulated in the SARS-CoV-2 Nsp6 transgenic fly hearts, and Fig.2. describes the analysis of a whole-genome RNA seq data. Fig 2A and Supplemental Table S1 summarize the gene's expression level changes upon Nsp6 over-expression. The authors mainly focus on genes that are relevant to the glycolysis pathway. Fig 2A chart cuts the fold differences to log2 from -3 to 3, ignoring the most significantly affected genes (Sup Table S1). I would recommend that Supplemental Table S1 data will be sorted based on Log2FoldChange value rather than on P-value. Also, in this section, the authors do not describe downregulated genes at all. Overall, the description of this very important data is biased toward the glycolysis pathway genes but should be more detailed and point to other significantly affected genes.

4. Line 348 section. Inhibition of glycolysis pathway activity by 2-deoxy-D-glucose (2DG) attenuates SARS-CoV-2 Nsp6-induced heart morphological and functional defects. Very nice demonstration of using *Drosophila* model for drug discovery. However, before assessing the drug's therapeutic effect, the authors should perform a thorough toxicity test and determine the maximum applicable dose of 2DR w/o detrimental effects. Next, the drug effects need to be shown as a dosage curve including 3-4 gradually increasing drug concentrations.

5. The analyses of the heart function are based on OCT imaging. The video recordings are very short, only 2 sec long. The heart rate variations of the same animal do occur; the length of the videos should be increased to 10 sec or longer.

6. The text contains a number of scattered typos. Please proofread and correct.

Reviewer #2 (Remarks to the Author):

The manuscript by Zhu et al. reports a model explaining the molecular basis of the cardiac damage caused by SARS-CoV-2 infection. According to this model, NSP6 is the principal protein involved. Experimental data show that the main effect of NSP6 is upregulation of glycolysis. This effect has already been described but here authors delineate the underlying molecular mechanism. The paper is clear, well written, the experimental work rigorous and will likely be influential in the field. Data treatment is appropriate.

A few questions:

I wonder whether there are indications that NSP6 variants may produce cardiac damage of different entity.

NSP6 is deemed to inhibit IFN-1 response and induce inflammatory cell death in lung epithelial cells: have these effects any mechanism or component in common with the here proposed glycolysis upregulation?

Responses to the reviewers' comments:

Reviewer #1:

The presented manuscript investigates the mechanisms of SARS-CoV-2 infection and describes possible therapeutic treatments. The authors build their conclusions on the results obtained from various model systems and research areas, demonstrating their broad scientific approach. Using Drosophila to study the detrimental effect of SARS-CoV-2 proteins represents a very efficient way to model the host-virus interactions. Overall, the manuscript represents a very detailed study of COVID19 disease consequences. The data and the discussion reflect the complexity of various symptoms and offer a strong hypothesis of the mechanism of cardiovascular complications. The manuscript can be accepted for publication after the authors address the following comments:

We very much appreciate the reviewer's positive response and their suggestions which have further strengthened our manuscript. We have incorporated all of these in the revised manuscript, details can be found below.

1. The authors use UAS/GAL4 system to express the viral genes specifically in the heart tissue. This is a legitimate way to introduce and study the effects of viral proteins on the tissue of interest. Hand-GAL4 drives the viral gene expression starting from the early stages of the development to adulthood, therefore the heart tissue develops with the viral protein being present. However, viral infection occurs when the tissue is completely formed. This difference between real viral infection and the presented model of viral infection should be acknowledged in the introduction, or at the beginning of the Results section.

We appreciate the reviewer making this important point. We have added this context "the observed heart phenotypes are due to the toxicity of SARS-CoV-2 genes during heart development" at the start of the Results section (p. 7).

2. Lower panel Fig 1D does not have a scale bar.

The scale bar for the lower panel in Figure 1D has been added, and the figure legend has been updated accordingly (p. 43). Furthermore, we noticed additional omitted scale bars in Figure 3B and Figure 5B; these have been added in the respective Figures and legends (p. 47 and 51). Our apologies for the initial oversight.

3. Line 201 section, Glycolysis genes are upregulated in the SARS-CoV-2 Nsp6 transgenic fly hearts, and Fig.2. describes the analysis of a whole-genome RNA seq data. Fig 2A and Supplemental Table S1 summarize the gene's expression level changes upon Nsp6 over-expression. The authors mainly focus on genes that are relevant to the glycolysis pathway. Fig 2A chart cuts the fold differences to log2 from -3 to 3, ignoring the most significantly affected genes (Sup Table S1). I would recommend that Supplemental Table S1 data will be sorted based on Log2FoldChange value rather than on P-value.

We concur data by Log2FoldChange rather than P-value would be beneficial to the readers and make it easier to identify additional genes that fall outside the dimensions of Figure 2A. Supplemental Table S1 is now provided with data sorted based on the Log2FoldChange value.

Also, in this section, the authors do not describe downregulated genes at all.

We have included analysis of the downregulated genes linked to the expression of SARS-CoV-2 Nsp6. The results for the Gene Ontology (GO) terms associated with the downregulated genes can be found in the new Supplemental Figure S2. Among the top terms was ribosome biogenesis (adj. $P = 1.05e-52$). Previously SARS-CoV-2 has been shown to bind the ribosome to disrupt protein translation (Banerjee et al., 2020). Moreover, downregulation of genes associated with ribosome biogenesis have been identified in HIV infection (Kleinman et al., 2014). These new results have been added to the

Results section (p. 9)

Overall, the description of this very important data is biased toward the glycolysis pathway genes but should be more detailed and point to other significantly affected genes.

To clarify, the analysis of the RNAseq data was unbiased; it revealed the upregulation of glycolysis genes. We focused on the glycolysis pathways because it was a top result in our GO analysis (see Figures 2D and 2E).

4. Line 348 section. Inhibition of glycolysis pathway activity by 2-deoxy-D-glucose (2DG) attenuates SARS-CoV-2 Nsp6-induced heart morphological and functional defects. Very nice demonstration of using Drosophila model for drug discovery. However, before assessing the drug's therapeutic effect, the authors should perform a thorough toxicity test and determine the maximum applicable dose of 2DR w/o detrimental effects. Next, the drug effects need to be shown as a dosage curve including 3-4 gradually increasing drug concentrations.

Thank you for this suggestion. We have added a Supplemental Figure S3 with dose-lethality curve for 2DG based on 0, 2, 10, 30, 50, 75, and 100 mM concentrations to improve the robustness of our data. Initially, we had investigated 0, 2, 10, and 50 mM 2DG treatments and found that 50 mM 2DG caused ~80% lethality in wild-type *Drosophila*, and 100% lethality in Nsp6 transgenic flies (p.16). The increased concentration, 75 and 100 mM 2DG, both caused (near) 100% lethality in the wild-type flies, thus informing the maximum applicable treatment dose in *Drosophila*. Whereas 30 mM 2DG resulted in ~35% lethality in wild-type flies. Altogether these data provide a smooth dose-response curve. These results can be found in the new Supplemental Figure S3 with accompanying description in the Results section (p. 15-16).

5. The analyses of the heart function are based on OCT imaging. The video recordings are very short, only 2 sec long. The heart rate variations of the same

animal do occur; the length of the videos should be increased to 10 sec or longer.

We appreciate the suggestion to strengthen our data. Two Supplemental Videos, each 15 seconds in length, with recording of the *Drosophila* heartbeat for the wild-type and Nsp6 flies have been added. Furthermore, we have used these to update the quantitation for the results in Figures 1, 3, and 5. The new length of the recorded videos has been updated in the Methods (p. 28).

6. The text contains a number of scattered typos. Please proofread and correct.

Our apologies for the typos. We have carefully read through the manuscript and captured & corrected as many typos and other language/grammatical errors as we could find. Among other we have corrected, “thos” to “those” (p. 18), “cells” to “cell” (p. 21), and “braking” to “breaking” (p. 32).

Reviewer #2:

The manuscript by Zhu et al. reports a model explaining the molecular basis of the cardiac damage caused by SARS-CoV-2 infection. According to this model, NSP6 is the principal protein involved. Experimental data show that the main effect of NSP6 is upregulation of glycolysis. This effect has already been described but here authors delineate the underlying molecular mechanism.

The paper is clear, well written, the experimental work rigorous and will likely be influential in the field. Data treatment is appropriate.

We very much appreciate the reviewers' positive comments and the thoughtful questions. We have addressed these below.

A few questions:

1. I wonder whether there are indications that NSP6 variants may produce cardiac damage of different entity.

It appears that infection with SARS-CoV-2 Omicron causes less severe disease than with prior variants. The contribution of prior infections and vaccination to this remain to be determined. That being said, several mutations have been identified in Nsp6 Omicron: Subtype BA.1 carries Nsp6 variants Δ 105-107 and I189V, whereas the current dominant variant, BA.5, carries Nsp6 variant Δ 106-108. It is possible that the Nsp6 variants affects its binding to the host MGA, PCGF6 and TFDP1 transcription factors thus limiting its ability to inhibit the formation of the MGA/MAX complex and rendering it incapable of shifting the balance towards MYC/MAX. With minimal increased glycolysis activity, the Nsp6 Omicron variants would be less toxic. This remains speculative. However, one of the advantages of the fly system is its rapid ability to model new variants. Their pathogenicity can then be compared across variants for multiple relevant tissues. We have added this notion to the discussion (p. 19-20).

2. NSP6 is deemed to inhibit IFN-1 response and induce inflammatory cell death in lung epithelial cells: have these effects any mechanism or component in common with the here proposed glycolysis upregulation?

An interesting notion. A previous study found that SARS-CoV-2 Nsp6 protein—together with Nsp13 and Orf6—can inhibit TBK1 and IRF3 activation, thus antagonizing Type 1 interferon (IFN-1) signaling (Xia *et al.*, 2020). IFN-1 signaling has been correlated with decreased glycolysis and mitochondrial damage (Olson *et al.*, 2021). Once IFN-1 signaling was inhibited, it induced glycolysis activity and caused tissue damage. However, while the previous study and our current findings both demonstrate that Nsp6 expression can induce glycolysis activity; the mechanisms causing the increased glycolysis activity were different. In our study, binding of SARS-CoV-2 Nsp6 to the MGA, PCGF6, and TFDP1 transcription factors inhibits the formation of the MGA/MAX complex, which shifts the balance towards MYC/MAX, thus leading to increased glycolysis activity.

REVIEWERS' COMMENTS:

Reviewer #1 (Remarks to the Author):

Great job with the revision!

Reviewer #2 (Remarks to the Author):

The authors have responded appropriately to the referee's comments, have improved their manuscript accordingly and strengthened their statements. In my opinion, the manuscript can be accepted for publication.

REVIEWERS' COMMENTS:

Reviewer #1 (Remarks to the Author):

Great job with the revision!

Reviewer #2 (Remarks to the Author):

The authors have responded appropriately to the referee's comments, have improved their manuscript accordingly and strengthened their statements. In my opinion, the manuscript can be accepted for publication.

Response: We appreciate the reviewers' time and their positive response to our research findings.
Thank you!